# Efficient and Selective Oxygenation of Cycloalkanes and Alkyl Aromatics with Oxygen through Synergistic Catalysis of Bimetallic Active Centers in Two-Dimensional Metal-Organic Frameworks Based on Metalloporphyrins

**DOI:** 10.3390/biomimetics8030325

**Published:** 2023-07-21

**Authors:** Xin-Yan Zhou, Bo Fu, Wen-Dong Jin, Xiong Wang, Ke-Ke Wang, Mei Wang, Yuan-Bin She, Hai-Min Shen

**Affiliations:** College of Chemical Engineering, Zhejiang University of Technology, Hangzhou 310014, China; 17816876326@163.com (X.-Y.Z.); fubo13479347998@163.com (B.F.); jwd2199484383@163.com (W.-D.J.); njy970615@163.com (X.W.); wangkk@zjut.edu.cn (K.-K.W.); sia2016_zhaohui@126.com (M.W.); shenhm2019@163.com (Y.-B.S.)

**Keywords:** C–H bonds, oxygen, partial oxygenation, MOFs, metalloporphyrins

## Abstract

Confined catalytic realms and synergistic catalysis sites were constructed using bimetallic active centers in two-dimensional metal-organic frameworks (MOFs) to achieve highly selective oxygenation of cycloalkanes and alkyl aromatics with oxygen towards partly oxygenated products. Every necessary characterization was carried out for all the two-dimensional MOFs. The selective oxygenation of cycloalkanes and alkyl aromatics with oxygen was accomplished with exceptional catalytic performance using two-dimensional MOF Co-TCPPNi as a catalyst. Employing Co-TCPPNi as a catalyst, both the conversion and selectivity were improved for all the hydrocarbons investigated. Less disordered autoxidation at mild conditions, inhibited free-radical diffusion by confined catalytic realms, and synergistic C–H bond oxygenation catalyzed by second metal center Ni employing oxygenation intermediate R–OOH as oxidant were the factors for the satisfying result of Co-TCPPNi as a catalyst. When homogeneous metalloporphyrin T(4-COOCH_3_)PPCo was replaced by Co-TCPPNi, the conversion in cyclohexane oxygenation was enhanced from 4.4% to 5.6%, and the selectivity of partly oxygenated products increased from 85.4% to 92.9%. The synergistic catalytic mechanisms were studied using EPR research, and a catalysis model was obtained for the oxygenation of C–H bonds with O_2_. This research offered a novel and essential reference for both the efficient and selective oxygenation of C–H bonds and other key chemical reactions involving free radicals.

## 1. Introduction

The oxygenation of hydrocarbons to produce usable oxygenated molecules is the potential strategy for maximizing the use of fossil resources [1,2,3,4]. In the current industry, widely accessible hydrocarbons may be converted into several key industrial chemicals or fundamental industrial raw materials through the shortest transformation path because of oxidative functionalization, in which partly oxygenated products (alcohols, aldehydes, and ketones), acids, and peroxides are produced [5,6,7,8,9]. When compared with other oxidants including *m*-chloroperoxybenzoic acid (*m*-CPBA) [6,10], hydrogen peroxide (H_2_O_2_) [11,12], iodosylbenzene (PhIO) [13,14], *tert*-butyl hydroperoxide (TBHP) [15,16,17], ozone (O_3_) [18] and sodium periodate (NaIO_4_) [19], molecular oxygen (O_2_) is recognized as the most attractive [20], because water (H_2_O) is the major by-product in the reduction of O_2_, while oxygen can be easily removed after the reaction. At the same time, the use of oxygen as an oxidant has the minimum oxidant cost and maximum atom economy. As a result, most commercialized oxygenated compounds are now produced by oxidizing hydrocarbons with oxygen. However, due to the inertness of O=O double bonds in molecular oxygen (bond dissociation energy is about 498 kJ/mol), and C–H bonds in hydrocarbons (bond dissociation energy is about 420 kJ/mol) [1,21,22,23,24], severe conditions, such as high temperatures, powerful oxidants, and corrosive additives, are normally required. For instance, in the commercial production of KA oil (cyclohexanone and cyclohexanone), the reaction temperature must approach 160–170 °C under a pressure of 1.0–2.0 MPa [25,26,27,28]. Another example is toluene and its derivatives, which must be oxidized using hazardous bromide additives and caustic acidic circumstances [29,30,31]. Because the produced oxygenated products have higher reactivity, the resulting oxygenated compounds can easily be excessively oxidized under such extreme conditions. Thus, the hydrocarbon conversion is generally kept at a lower level to achieve a decent selectivity toward partly oxygenated products, resulting in low reaction efficiency. In commercial cyclohexane oxygenation, conversion is generally regulated at around 5% to offer a higher selectivity (75–85%) towards KA oil [26,32,33,34]. It is a critical necessity to minimize deep oxygenation and selectivity loss with conversion growth in commercial cycloalkane oxygenation and alkyl aromatic oxygenation with oxygen.

Because the oxygenation of C–H bonds with oxygen is mainly a radical process and the partly oxygenated products are in an environment with many active free radicals, they might be readily attacked and over-oxidized, resulting in the loss of selectivity with increased substrate conversion. Second, the main oxygenation intermediates, alkyl hydroperoxides, are chiefly transformed into ketones and alcohols through thermal disintegration, which proceeds through a chaotic radical path, raises the number of free radicals in the catalytic system, and causes the ketones and alcohols to be deeply oxygenated [35,36,37]. Third, in a homogeneous oxygenation system, frequent interaction between partly oxygenated products and catalytic active centers is unavoidable, resulting in deep oxygenation. Consequently, to successfully avoid deep oxygenation, obtain a satisfying conversion, and achieve high selectivity, the factors indicated above should be applied strategically in the oxygenation of C–H bonds with oxygen.

In hydrocarbon oxygenation with oxygen as an oxidant, many novel innovations have been documented to efficiently inhibit deep oxygenation from enhancing conversion and selectivity. Among these innovations, photocatalysis is regarded as a highly successful method that may be performed at room temperature [15,38,39]. Deep oxygenation occurring to the oxygenation products can be effectively prevented at lower reaction temperatures. For instance, Tüysüz and colleagues used bismuth halide perovskite in SBA-15 silica as a catalyst in the oxygenation of C–H bonds at room temperature with solar light illumination employing oxygen [38]. Utilizing toluene as a substrate allowed for a 90% selectivity in producing the partially oxygenated product aldehyde, thanks to the mild reaction temperature and the effective prevention of extensive oxygenation. Shao and co-workers demonstrated an effective suppression of deep oxygenation by minimizing the contact of partly oxygenated products with a catalytically active center, in addition to lowering the reaction temperature, in which a cobalt-containing ZSM-5 molecular sieve was modified with ionic liquid, and the catalytic material generated was employed to cyclohexane oxygenation with oxygen [26]. Because partly oxygenated products are more soluble in ionic liquid, the interaction of partly oxygenated products with catalytically active core cobalt is considerably decreased, preventing deep oxygenation. The KA oil selectivity was enhanced from 73% to 92% [26].

Some of our research on C–H bond oxygenation has used bimetallic catalytic devices to catalyze the conversion of oxygenation intermediate R–OOH and limit disordered thermal disintegration of R–OOH to lower deep oxygenation [40,41,42]. Deep oxygenation was considerably avoided by the catalytic conversion of cycloalkyl hydroperoxides. In a system with bimetallic catalysts for the oxygenation of cyclohexane, the total selectivity to KA oil increased from 80% to 97%, and the conversion of the substrate was enhanced from 3.8% to 6.5% [42]. Taking inspiration from the aforementioned studies, it could be a practical strategy to limit the deep oxygenation of partly oxygenated products by reducing the reaction temperature, preventing partial oxygenation products from coming into contact with the catalytic active center and catalytic conversion of intermediate alkyl hydroperoxides to avoid selectivity decrease. If integrating the above techniques into one catalytic system, it might be a very effective catalytic strategy in the industrial oxygenation of C–H bonds selectively, which are rarely described.

Metal-organic frameworks (MOFs) are porous materials composed of metal nodes and multi-site organic linkers connected by coordination bonds to form a periodic network structure [43,44,45,46,47,48,49,50]. The characteristic structure of MOFs can be used to build catalytic systems with multi-metal active centers because metal active centers may be distributed in both the metal nodes and organic linkages via coordination bonds at the atomic level. Furthermore, the porous structures of MOFs can suppress the chaotic diffusion of the reactive intermediate and be utilized as confined catalytic realms [51,52,53,54,55], which can avoid oxidation by adjusting the disordered transport of free radicals. When applied in hydrocarbon oxygenation by oxygen, multiphase catalytic material MOFs can minimize the contact area of oxygen-containing products with catalytic active centers, therefore inhibiting deep oxygenation and enhancing conversion and selectivity. For this reason, in this work, two-dimensional MOFs with double metal cores (Co and Ni, Co and Fe, Co and Mn) were constructed using metalloporphyrins (tetrakis(4-carboxylphenyl)porphyrin metal(II), abbreviated as TCPPM or T(4-COOH)PPM) as organic linkers, as shown in Figure 1 [56,57,58,59,60] to suppress deep oxygenation during C–H bond oxygenation. As a continuation and extension of our prior work, the pores in two-dimensional MOFs were exploited to govern chaotic radical transport under mild conditions (120 °C) because they can confine free radicals in fixed realms [40,41,42,61,62,63,64,65]. In the obtained catalytic systems, cobalt was used as a catalytically active center to oxygenate C–H bonds to R–OOH with oxygen.

By contrast, Ni or Fe were used as catalytically active centers to facilitate the catalytic conversion of alkyl hydroperoxides. In addition, Ni or Fe have been proven to be inefficient in the catalytic oxygenation of hydrocarbons by oxygen directly in our work. The suppression of free-radical diffusion, the reduced contact of partial oxygen-containing products with the catalytically active center, and the mild reaction temperature (120 °C) all resulted in the significant suppression of the deep oxygenation of hydrocarbons with oxygen. In addition, the double metal centers were used to convert oxygenation intermediates alkyl hydroperoxides catalytically to avoid a selectivity decrease. When cyclohexane oxygenation was used as a model reaction, the selectivity to partly oxygenated products increased from 85.4% to 92.9%, and conversion increased from 4.4% to 5.3% when compared to sole metalloporphyrin (T(4-COOCH_3_)PPCo) as a catalyst. In addition, experimental research, electron paramagnetic resonance (EPR) analyses, and radical trapping experiments were used to investigate the synergistic catalysis model and oxygenation process of hydrocarbon in this work. As we know, this work presented here is an advanced, novel, and pragmatic example of C–H bond oxygenation. In this research, not only is the deep oxygenation greatly inhibited, satisfying selectivity to the partly oxygenated products being obtained, but also hydrocarbon conversion is simultaneously improved by a comprehensive and rational regulation of the reaction path. This work was insightful and beneficial both in commercial applications involved in selective oxygenation with oxygen, and academic research about other essential chemical conversions.

## 2. Experimental Section

The Appendix A contains information on the chemicals used, materials, characterizations, instrumentations, and syntheses of metalloporphyrins, as well as research related to free-radical traps.

### 2.1. Syntheses of Two-Dimensional MOF M1-TCPPM2

A catalyst based on two-dimensional MOF (M_1_-TCPPM_2_) (M_1_=Co(II), Cu(II), Zn(II), Mn(II), Ni(II), M_2_=Co(II), Cu(II), Zn(II), Mn(II), Fe(II), Ni(II)) was synthesized by the method shown in Figure 1 with some changes to the procedure in the literature [56,58,60]. Tetrakis(4-methoxycarbonylphenyl)porphyrin metal(II) T(4-COOH)PPM_2_ (TCPPM_2_, 0.10 mmol) was mixed with 30 mL DMF under a nitrogen environment in a reaction tube (120 mL), followed by the addition of metal nitrate (M_1_NO_3_) (1.20 mmol) in 4 mL HNO_3_ (1.0 mol/L). The resultant mixture was ultrasonically mixed for 0.5 h. Then the reaction was carried out at 90 °C for 72.0 h under nitrogen protection. After the precipitate was gathered at the ambient temperature, it was washed five times with dry N,N-dimethylformamide (5 × 6.0 mL), and acetone (5 × 6.0 mL). Dark red powder materials, such as Co-TCPPNi(ICP-MS: Co, 2.91%. Ni, 2.63%. m/m), Co-TCPPFe(ICP-MS: Co, 1.97%. Fe, 2.15%. m/m), Co-TCPPMn, Mn-TCPPZn, Mn-TCPPFe, Mn-TCPPNi, were obtained by drying the resulting red solid under decreased pressure at 70 °C for 12.0 h. The same process was used to synthesize Co-TCPPCo, Mn-TCPPMn, and Ni-TCPPNi with a monometallic center.

### 2.2. Catalytic C–H Bonds Oxygenation with Oxygen

The substrates cyclohexane (200 mmol) and two-dimensional MOF catalyst were charged in a reactor (100 mL high-pressure reactor equipped with a magnetic stirrer and a Teflon lining). After stirring, it was heated to the desired temperature in a silicon oil bath. The reactor was filled with molecular oxygen (99.99% purity) after reaching the desired temperature. After stirring for a specified reaction time at the set temperature and with the reaction temperature monitored continuously, the reactor was cooled to room temperature with cool water; then, triphenylphosphine was added to convert the remaining peroxide and the resulting liquid was accurately diluted with acetone to 100 mL by stirring for 30 min. To determine conversion and selectivity, the resultant solution was submitted to gas chromatography using 10 mL with toluene as the internal standard. Similarly, 10 mL of the solution of acetone was mixed with benzoic acid (internal standard) to conduct HPLC analyses. All the products were identified qualitatively by comparing them to the standard samples in chromatographic analyses. The oxygenation products were examined using a Thermo Trace 1300 gas chromatographer equipped with a Flame Ionization Detector and a TG-5MS capillary column (30 m × 0.32 mm × 0.25 μm). A Thermo Ultimate 3000 HPLC chromatographer with Photodiode Array Detector and Amethyst C18-H liquid chromatography column (250 mm × 4.6 mm × 0.25 μm) was used to examine acids. The internal standard for GC analysis was toluene, while the internal standard for HPLC analyses was benzoic acid.

### 2.3. Apparent Kinetic Research

To verify why the prepared two-dimensional MOFs have various catalytic properties, apparent kinetic experiments were carried out under 120 °C, 115 °C, and 110 °C for the oxygenation of cyclohexane with Co-TCPPNi, Co-TCPPFe, and Co-TCPPMn as catalysts. Without a catalyst, cyclohexane was oxidized under 140 °C, 135 °C, and 130 °C. Under standard procedure 2.2, the catalyst and substrate were introduced into a reactor with a Teflon liner, with a volume of 100 mL. Oxygen was added to the reactor when the temperature reached a specified point. The reaction mixture was immediately cooled to 25 °C after stirring for 6.0, 6.5, 7.0, 7.5, and 8.0 h under the set condition, and triphenylphosphine was added to convert the remaining peroxide. Finally, the resulting liquid was accurately diluted with acetone to 100 mL.

### 2.4. Electron Paramagnetic Resonance (EPR) Analyses

Two-dimensional MOF Co-TCPPNi and substrate cyclohexane were charged in the reactor. Combined with stirring, the reactor was heated to 130 °C; then oxygen was injected into the reaction system until the pressure reached 1.0 MPa. A total of 1 mL of the resultant mixture was withdrawn, and then 5,5-dimethyl-1-pyrroline N-oxide (DMPO) was added to catch free radicals after stirring at a temperature of 130 °C and a pressure of 1.0 MPa for 6.0 h. The resultant solution was then submitted to EPR examination at room temperature using a JEOL JES-FA200 spectrometer.

## 3. Results and Discussion

### 3.1. Catalyst Characterization

To determine the microstructures of the two-dimensional MOFs with double metal catalytic centers prepared, several characterizations were used, including nitrogen adsorption–desorption isotherms (BET), scanning electron microscopy (SEM), transition electron microscopy (TEM), energy dispersive spectra (EDS), X-ray photoelectron spectra (XPS), Fourier transform infrared spectra (FT-IR), etc. Both metalloporphyrins and two-dimensional MOFs were characterized by FT-IR, which were illustrated in Figure 1 and Appendix A. The sharp peaks at approximately 1690 cm^−1^ (black dots) may be attributed to C=O bond-stretching vibrations, and the broad bands at roughly 3000 cm^−1^ and 3500 cm^−1^ were C–H and O–H bond-stretching vibrations. Because of the coordination between metal nodes and –COOH, the distinctive vibrations of C=O bonds in MOFs vanished or became not apparent as compared to metalloporphyrins. The characteristic vibration of the benzene skeleton was a strong peak near 1600 cm^−1^ (green dot). The characteristic vibrational peaks of metalloporphyrin at 1695 cm^−1^ (red point), 1600 cm^−1^ (green point), 1400 cm^−1^ (black point), and 1000 cm^−1^ (blue point) were preserved in the synthesized two-dimensional MOFs, demonstrating the correct structure of the prepared two-dimensional MOFs (Figure 1 and Appendix A). In addition to the FT-IR, XPS analysis was used to determine the successful syntheses and element components of the prepared two-dimensional MOFs, as shown in Figure 2. All the characteristic metallic elements were identified by the XPS analysis of the prepared two-dimensional MOFs (Figure 2a). In particular, in the MOFs with double metal catalytic centers, two metallic elements were found. Peaks at approximately 801 and 785 eV in the XPS survey spectra (Figure 2b–e) may be ascribed to the Co 2p_1/2_ and Co 2p_3/2_ [66,67,68], 877 and 860 eV to Ni 2p_1/2_ and Ni 2p_3/2_ [69,70,71], and 656 and 643 eV to Mn 2p_1/2_ and Mn 2p_3/2_ [72,73,74], respectively. Co^2+^, Co^3+^, and satellite peaks are the three portions of the Co 2p composition in two-dimensional MOFs, with Co^2+^ and Co^3+^ existing in a molar ratio of 3:1. In the case of Mn, it was mostly found at +2 valence and Ni was all in +2 valence. As a result, the valences of the catalytical active centers in the two-dimensional MOFs with double metal catalytic centers prepared in this study were mostly +2 and +3. PXRD was used to determine the crystallinity of the prepared two-dimensional MOFs, and the obtained spectra were demonstrated in Figure 3. The discrepancy in the PXRD pattern between the synthesized and the simulated two-dimensional MOFs, as shown in Figure 3, was mostly due to the stacking of two-dimensional lamellar MOFs in the irregular state and the formation of other metal clusters that differ from the binuclear clusters, as indicated in Figure 1.

The geometrical shape and microstructure of the prepared two-dimensional MOFs were subsequently investigated using SEM, TEM, and BET. Figure 4 shows the SEM images of the catalysts at various magnifications. It was revealed that the prepared two-dimensional MOFs were mostly flaky structures, and obvious lamellar structure could be observed, which stacked into the uniform flaky structure of the two-dimensional MOFs (Figure 4). TEM measurements were also used to verify the flaky structure of the prepared two-dimensional MOFs [67,72,75], as well as the stacking morphology of their lamellar structures, as shown in Figure 5. Not only are the flaky structures (Figure 5a,b,d,e,g,h) shown in the TEM images, but also element distributions were obtained (Figure 5c,f,i) [73,76,77]. All fundamental elements and associated metals were found in the EDS mapping analyses along with TEM examinations, and their distributions in the synthesized MOFs were remarkably homogeneous. SEM, TEM, and EDS tests gave convincing evidence that the two-dimensional MOFs were successfully constructed. The non-local density functional theory (NLDFT) model of nitrogen adsorption–desorption isotherms was then used to determine the pore characteristics of the prepared two-dimensional MOFs. As illustrated in Figure 6 and Appendix A, all the two-dimensional MOFs used in this work possessed micropore structures (pore widths smaller than 2.0 nm), which may provide sufficient confined realms for the C–H bond oxygenation employing oxygen to limit chaotic radical transport. Micropores with pore widths of 1.09 nm were detected in the two-dimensional MOF Co-TCPPNi, and the BET-specific surface area was 324.1 m^2^/g with a pore volume of 0.042 cm^3^/g, which was useful for the substrate to enter the micropores. In addition, the presence of micropores in Co-TCPPNi may be responsible for its superior catalytic performance compared to materials, because the micropores could prevent chaotic radical transport, and the abundant mesopores in Co-TCPPNi could increase the substrate accessibility to the active centers.

Finally, TGA measurements in the air were carried out to determine the thermostability of the prepared two-dimensional MOFs as catalytic materials in oxygenation reactions. Figure 7 shows that the weight reduction of catalysts happened primarily above 240 °C. The reaction temperature of C–H bond oxygenation employing oxygen in this research was far below 240 °C. It is possible to deduce that the two-dimensional MOFs in this work might meet the thermal stability requirement of hydrocarbon oxygenation and could act as an effective catalyst.

As a summary of catalyst characterizations, the catalysts prepared in this study were microporous lamellar structures, which were stacked into uniform flaky structures with double metal catalytic centers. The metal centers existed homogeneously in the obtained two-dimensional MOFs in +2 and +3 valences. Under air, the thermodynamic breakdown temperature of the prepared catalysts might reach up to 240 °C, which was higher than the C–H bond oxygenation reaction temperature (below 140 °C). The pores in Co-TCPPNi offered sufficient confined realms to prevent chaotic free-radical transport, resulting in enhanced catalytic performance in the partial oxygenation of cycloalkanes and alkyl aromatics with oxygen.

### 3.2. Preliminary Experiments for Cyclohexane Oxygenation as Model Reaction

Before the comprehensive research of the C–H bond oxygenation catalyzed by two-dimensional MOFs with double metal catalytic centers, some fundamental experiments were conducted with cyclohexane oxygenation as a model reaction. The fundamental problem was to determine an appropriate reaction temperature. To achieve precise and efficient C–H bond oxygenation with oxygen, the temperature of the reaction should be set at a temperature range that could prevent apparent autoxidation, which would result in the deep oxygenation of partly oxygenated reactive products, lowering the selectivity and increasing the uncertainty of the oxygenation process. Appendix A illustrates that the apparent autoxidation of cyclohexane, which is essential in the chemical industry, happened primarily at 135 °C, at which the conversion was as high as 1% (Entry 8 in Appendix A). At 120 °C or lower, the conversion of cyclohexane was less than 0.15%. To avoid non-selective and uncontrolled autoxidation, the reaction temperature of cyclohexane in this study was set to 120 °C, at which no significant autoxidation occurred. Then, the feasibility of precise and efficient C–H bond oxygenation catalyzed by double metal catalytic centers was investigated as the appropriate reaction temperature was determined. No appreciable autoxidation happened at 120 °C and no catalytic performance was found at 120 °C with porphyrin ligand (T(4-COOCH_3_)PP (Porp.)) without metal as a catalyst, as shown in Table 1 (Entry 1, Entry 2). Using T(4-COOCH_3_)PPCo (Porp.Co), the catalytic oxygenation of cyclohexane went smoothly, with conversion and selectivity for partly oxygenated products being 4.4% and 85.4%, respectively (Entry 3 in Table 1). Even though there was no significant catalytic performance obtained when T(4-COOCH_3_)PPNi (Porp.Ni) or T(4-COOCH_3_)PPFe (Porp.Fe) was used as catalyst, there was a little increase in conversion and selectivity to partly oxygenated products when two metalloporphyrins Co(II) and Ni(II) were used as double metal catalytic system (Entry 8 in Table 1). The satisfying conversion and selectivity also implied the feasibility of the double metal catalytic system in C–H bond oxygenation with oxygen. When a two-dimensional metalloporphyrin-based MOF with double metal catalytic centers, Co-TCPPNi, was utilized as a catalyst, a boost from 4.6% to 5.6% was obtained in the cyclohexane conversion with no appreciable selective depletion (Entry 10 in Table 1). In comparison with metalloporphyrin T(4-COOCH_3_)PPCo (Porp.Co), the use of Co-TCPPNi with double metal catalytic centers as catalysts increased the conversion of cyclohexane with good selectivity as well as the catalytic system of T(4-COOCH_3_)PPCo (Porp.Co) and T(4-COOCH_3_)PPNi (Porp.Ni). The successful use of two-dimensional metalloporphyrin-based MOFs Co-TCPPNi with double metal catalytic centers as catalysts and a satisfying catalytic performance strongly supported the feasibility of precise and efficient oxygenation of cycloalkanes and alkyl aromatics with oxygen in the presence of two-dimensional metalloporphyrin-based MOFs with double metal catalytic centers as catalysts.

### 3.3. Synergistic Catalytic Oxygenation of C–H Bonds with M_1_-TCPPM_2_

After establishing the feasibility of pursuing the precise and efficient oxygenation of C–H bonds through catalysis of double metal catalytic centers, a variety of two-dimensional MOFs with double metal catalytic centers were prepared and tested in C–H bond oxygenation with cyclohexane as a model substrate. For comparison, the comparable two-dimensional MOFs with single metal cores were also prepared. Table 2 and Appendix A summarized their performances in the catalytic oxygenation of cyclohexane with oxygen. When compared to T(4-COOCH_3_)PPCo (Porp.Co), Co-TCPPCo with only one metal center performed a little better in the catalytic experiments. Conversion climbed to 4.5% from 4.4%, while selectivity increased from 85.4% to 86.3%.

Furthermore, M_cyclohexanol_:M_cyclohexanone_ (molar ratio) was raised from 1:1 (Entry 11 in Table 2) to 1.1:1. The increase in M_cyclohexanol_:M_cyclohexanone_ (molar ratio) is a strong indicator of the OH· rebounding mechanism and suppression on the chaotic free-radical spread [78,79,80,81], which is one of the initial goals of conducting C–H bond oxygenation in porous MOFs. Because of the confined catalytic realms of two-dimensional MOFs, chaotic free-radical transport was reduced, which was compatible with our initial goal in this study. No products were detected in the utilization of two-dimensional MOFs with sole metal center Ni (Ni-TCPPNi) as a catalyst. Then, the catalytic properties of two-dimensional MOFs with double metal catalytic centers were thoroughly explored with the achievement of the inhibition of chaotic free-radical transport in two-dimensional MOFs. Table 2 demonstrates that two-dimensional MOFs with Co(II) as a metal node outperformed in C–H bond oxygenation employing cyclohexane as a model compound compared to other metal nodes. The conversion of cyclohexane in Co-TCPPNi catalysis was up to 5.6%, with a selectivity of 92.9% towards KA oil (Entry 12 in Table 2). The results were better than both datasets of homogeneous metalloporphyrin catalysis (Entry 13 and Entry 14 in Table 2) and the results from using the physical mixture of two metalloporphyrins as catalysts (Entry 15 in Table 2). In the oxygenation of hydrocarbons with oxygen, the simultaneous improvement of conversion and selectivity were achieved in our work, which was very appealing, because, in most hydrocarbon oxygenations with oxygen, the selectivity for partly oxygenated products would be depleted when conversion increased.

The improved selectivity for partly oxygenated products in the Co-TCPPNi catalysis of cyclohexane oxygenation is mainly attributed to the inhibition of radical transport by the confined catalytic realm of Co-TCPPNi and the reduction of partly oxygenated products contacting the active catalytic center during multiphase catalysis in this study. The catalytic conversion of the oxygenation intermediates peroxides, which effectively inhibited the chaotic thermal degradation of R–OOH and enhanced the innate oxygenation ability of R–OOH in the use of bimetallic active centers, is greatly responsible for the superior selectivity too, which also was the reason for the superior conversion in Co-TCPPNi catalysis of cyclohexane oxygenation. Experiments confirmed that the bimetallic active centers increased the oxygenation ability of alkyl hydroperoxides. No oxygenation product identified as cyclooctane was combined with oxidative cyclohexyl hydroperoxide in cyclohexane without a catalyst, as shown in Appendix A. When T(4-COOH)PPNi, Co(OAc)_2_, T(4-COOCH_3_)PPNi, or their simple mixtures were utilized, KA oil was detected from the reaction products, despite the absence of catalytic characteristics in the Ni(II) catalysis of cyclohexane oxygenation with oxygen reported in Table 2. This finding suggests that the addition of Ni(II) can promote the employment of oxidative species (cyclohexyl hydroperoxide) as an extra oxidant, preventing its accumulation and disordered thermal disintegration, therefore improving the conversion and selectivity. As a phased summary, Figure 8 summarizes a synergistic oxygenation model based on the thorough investigation of the conversion process in the Co-TCPPNi catalysis of hydrocarbon oxygenation. To explain why the catalytic properties of two-dimensional MOFs with double metal catalytic centers were better than one metal catalytic center, we propose this model. Table 2 (Entry 2 and Entry 14) shows that only the Ni metal catalytic center had no catalytic reactivity. Metal catalytic centers Co and Ni performed better than only the Co metal catalytic center (Entry 10 and Entry 12 in Table 2). Otherwise, the catalytic system with Co and Ni could oxygenate cycloalkane efficiently by employing cyclohexyl hydroperoxide (Appendix A). According to these experiment results, we thought that the system employing two-dimensional MOFs with double metal catalytic centers has two phases. Hydrocarbons are oxidized by oxygen catalyzed by Co(II) to form alkyl hydroperoxides in the first phase. In the second phase, the resulting hydroperoxides were then used as extra oxidizing agents to convert new substrates to partly oxygenated products catalyzed by double metal catalytic centers or were decomposed catalytically by active centers Co(II) and Ni(II). An obvious synergistic catalysis model was observed from the double metal centers. Using synergistic catalysis, the catalytic conversion of reactive species peroxide and the full use of its intrinsic oxygenation properties were accomplished, leading to improved catalytic performance, which was the major reason for the better results of Co-TCPPNi in effective C–H bond oxygenation with oxygen. In addition, the synergistic oxygenation model also offers a potentially unique and practicable technique for C–H bond oxygenation with oxygen in the chemical industry.

### 3.4. Further Optimizing Reaction Conditions

When cyclohexane was the substrate, the reaction conditions of hydrocarbon oxygenation were further optimized using the optimum catalytic materials in terms of catalyst amount and pressure. As indicated in Table 3, when Co-TCPPMn, Co-TCPPNi, or Co-TCPPFe was utilized as an exemplary catalyst, the highest catalytic performance obtained as catalyst dosage is 0.08 mg/mmol (Entry 3, Entry 9, and Entry 15 in Table 3). As the catalyst amount increased further, there was a considerable drop in conversion, which might be attributed to the quenching action on the active radicals of the water adsorbed by the metallic nodes. The conversion of cyclohexane was practically constant under oxygen pressure from 0.60 MPa to 1.40 MPa, and no evident effect on the conversion was discovered by adjusting the oxygen pressure in Table 4. This is mostly because oxygen was provided continuously and kept in excess in our study. In this research, 1.0 MPa should be the appropriate pressure of oxygen. Through the previously mentioned optimizations, it was also obvious that Co-TCPPNi outperformed Co-TCPPFe and Co-TCPPMn in terms of catalytic performance. Systematic optimization of the experimental parameters was extremely recommended for precise and efficient C–H bond oxygenation utilizing cyclohexane as a model substrate. The two-dimensional MOF Co-TCPPNi was the selected catalyst based on these optimal circumstances. With a catalyst dosage of 0.08 mg/mmol, at the temperature of 120 °C and pressure of 1.0 MPa, after reacting 8.0 h, the conversion of cyclohexane was as high as 5.6% with a selectivity of 92.9% for partly oxygenated products under optimized reaction conditions. The result was better than the catalytic performance obtained when utilizing the corresponding homogeneous metalloporphyrin as a catalyst (4.4%, 85.4%), and its simple mixture (4.6%, 88.5%). In our efforts to recover the used catalyst, we encountered challenges due to the low catalyst loading (0.08 mg/mmol, catalyst/substrate) and the limited amount of catalyst used in each experiment (16.0 mg). Unfortunately, we were unable to successfully recover the catalyst Co-TCPPNi. However, in our future work, we remain committed to resolving this issue and achieving the recyclability of the synthesized catalysts.

### 3.5. Apparent Kinetics Estimation

The remarkable performance of Co-TCPPNi in C–H bond oxygenation with oxygen was investigated further by the means of apparent kinetic research, with cyclohexane acting as a model substrate and Co-TCPPFe, Co-TCPPMn serving as comparison catalysts. The apparent kinetic investigation adopted a continuous oxygen supply and a stirring velocity of 600 rpm. As shown in Figure 9 and Appendix A, and Table 5 and Appendix A, the pseudo-second-order kinetic model had the highest correlation coefficients among the pseudo-zero-order, pseudo-first-order, and pseudo-second-order kinetic models, when fitting the reaction data of substrate autoxidation and catalytic oxygenation with reaction times. As a result, it is reasonable to assume that C–H bond oxygenation with oxygen here was predominantly a pseudo-second-order kinetic process, with relational formula between conversion and reaction time as Equation (1).
*kt* = *X*_A_/[*C*_A0_·(1 − *X*_A_)] (1)

*X*_A_ represents the cyclohexane conversion rate in the rate equation above. *t* represents the time, *C*_A0_ represents the starting cyclohexane concentration and *k* is the reaction rate constant obtained in Table 5 for corresponding reaction temperatures. As *k* values at specific temperatures were obtained from pseudo-second-order kinetic fittings, the apparent activation energy (*Ea*) of the autoxidation and catalytic oxygenation in this work was calculated using the Arrhenius equation, as shown in Equation (2).
*lnk* = −(*Ea*/R) (1/*T*) + *lnk*_0_
(2)

Table 5 shows pseudo-second-order kinetic parameters for cyclohexane oxygenation, as well as corresponding Ea. The Ea of cyclohexane autoxidation was 163.9 kJ/mol. The Ea was 82.55 kJ/mol, 85.91 kJ/mol, and 133.08 kJ/mol as Co-TCPPNi, Co-TCPPFe, and Co-TCPPMn being used as catalysts. The capacity of catalysts to lower the apparent activation energy (Ea) was consistent with their performance in catalytic cyclohexane oxygenation. The increased catalytic performance meant lower apparent activation energy. The kinetic research above presents credible evidence for the various properties of catalysts in cyclohexane oxygenation, as well as a deeper understanding of the catalytic activity of two-dimensional MOFs with double metal catalytic centers in C–H bond oxygenation and the origin of their different catalytic properties.

### 3.6. The Systematic Investigation into Substrate Scope

An optimal technique for the synergistic catalysis of hydrocarbon oxygenation was developed using cyclohexane as a model substrate and Co-TCPPNi with the double metal catalytic center as a catalyst. In addition, several cycloalkanes were added to the list of substrates. The main oxygenation product was ketones and alcohols, which was in accordance with other references. As indicated in Table 6 and Table 7, the selectivity of partly oxygenated products could approach 92% or higher when Co-TCPPNi was used as the catalyst with excellent conversions to typical cycloalkanes. Furthermore, the conversion grew when the ring of cycloalkanes expanded. The selectivity of partly oxygenated products was 92.9% in cyclohexane oxygenation, which is an essential process in the petrochemical industry. In addition, the conversion of cyclohexane was 5.6%, which would be an appealing and valuable level for commercial cyclohexane oxygenation. The conversion increased to 30.7% when the substrate was changed to cyclododecane, with no deep oxygenation products being found. The substrate compatibility of the oxygenation system acquired in this study might be suitable to typical alkyl aromatics as substrates too, due to the high catalytic activity of Co-TCPPNi in cycloalkane oxygenation. Under solvent-free conditions, the predominant oxygenation products of secondary benzylic C–H bonds were acetophenone, and its derivatives with valuable 1-phenylethanol, 1-phenylethyl hydroperoxide, and its derivatives as byproducts. Conversion of 4-bromoethylbenzene was 59.2% with a selectivity of 97% for 4-bromoacetophenone. The main product of the oxygenation of isopropylbenzene and its derivatives was valuable peroxide, which was an essential intermediate and compound in the industrial oxygenation of alkenes. The systematic investigation into substrate scope revealed the significant potential and utility of the proposed strategy of synergistic catalysis using Co-TCPPNi as a catalyst under solvent-free and additive-free conditions. This approach proved highly effective in enhancing the yield of partially oxygenated products while maintaining a desirable selectivity for such products during C–H bond oxygenation.

### 3.7. Synergistic Mechanism

The mechanism of the synergistic catalysis of C–H bond oxygenation was then investigated using cyclohexane as a model substrate. Hydrocarbon oxygenation with oxygen or alkyl hydroperoxides is thought to occur mostly by a radical process [35,37,82]. The free-radical catcher CBrCl_3_ and Ph_2_NH were added to the cyclohexane oxygenation, employing Co-TCPPNi as a catalyst to determine whether C–H bond oxygenation proceeded through a radical process [20,83]. The conversion of cyclohexane was substantially reduced from 5.6% to 1.0% and 1.1% when CBrCl_3_ and Ph_2_NH were added at a 5% (molar ratio) to the catalytic oxygenation of cyclohexane, as shown in Appendix A. The apparent quenching effect provided strong support for the free-radical process. In the GC-MS analysis of quenching tests using CBrCl_3_ as a radical catcher, brominated cyclohexane product and chlorinated cyclohexane product were discovered, indicating the existence of cyclohexyl (C_6_H_11_·), as shown in Appendix A. The presence of hydroxyl radicals (HO·) was strongly substantiated by the discovery of (CH_3_)_3_COH in the GC-MS study when (CH_3_)_3_CBr was employed as a radical catcher (Appendix A). In addition, electron paramagnetic resonance (EPR) research was used to analyze the radical species in the C–H bond oxygenation in the catalysis of Co-TCPPNi. When 5,5-dimethyl-1-pyrroline-N-oxide (DMPO) was used as a spin trap in the reaction mixture, not only cyclohexyl radical (C_6_H_11_·) but also cyclohexyl peroxy radical (C_6_H_11_OO·) and cyclohexyl oxy radical (C_6_H_11_O·) were discovered (Figure 10). As a result of the analysis of radical species during the cyclohexane oxygenation with oxygen catalyzed by Co-TCPPNi, the reaction pathway of this oxygenation system was verified to be a radical process, with the detected radical intermediate primarily being cyclohexyl (C_6_H_11_·), hydroxyl (HO·), cyclohexyl peroxy radical (C_6_H_11_OO·), and cyclohexyl oxy radical (C_6_H_11_O·). The free-radical intermediates discovered above were consistent with those described in the literature [26,37,84].

The mechanism of C–H bond oxygenation through the synergistic catalysis of Co-TCPPNi is presented in Figure 2, with references to certain relevant studies [26,37,79,82,84,85,86,87,88] after determining the primary radical species, radical mechanism, and synergistic catalysis model. The oxygenation reaction pathway of hydrocarbons with oxygen catalyzed by Co-TCPPNi, as demonstrated in Figure 2, can be separated into four major sub-paths that are interconnected with each other. During the first sub-path (black arrow), Co(II) in Co-TCPPNi activated oxygen, resulting in the active center [Co(IV)=O], which may pull the H atom from the C–H bond to form the hydroxylated metal [Co(III)-OH] and the alkyl carbon center radical R·. The hydroxyl group in [Co(III)-OH] might rebound to R·, creating the corresponding alcohol R–OH, which could be over-oxidized to alkyl ketone and others. In this sub-path, most of the hydroxyl groups in [Co(III)-OH] could not rebound to R· and form R–OH as shown in the first sub-path during the reaction. The released R· reacted with oxygen to generate alkyl peroxy radicals, which could pull the H atom from the hydrocarbon and then generate R–OOH in the second sub-path (orange arrow). The generated alkyl hydroperoxide was used as an oxidant to convert the hydrocarbons to alcohols and ketones catalyzed by Ni(II), which could avoid the chaotic thermodynamic disintegration of alkyl hydroperoxide and result in the improvement of the selectivity in the hydrocarbon oxygenation. The third sub-path (pink arrow) uses Co(II) to speed up the hydrocarbon oxygenation with alkyl hydroperoxide as an oxidant. Both the second and third pathways are key points for efficient and controlled hydrocarbon oxygenation with oxygen, and they are the central phases of cycloalkanes and alkyl aromatic oxygenation with oxygen in the synergistic catalysis of Co-TCPPNi. The radical diffusion mechanism (blue arrow, forth sub-path), which is invariably present during the C–H bond oxygenation with oxygen, can yield both alcohols and ketones in a 1:1 molar ratio. The strategy we wanted to use to overcome the critical challenge of disordered radical transport in this work was the use of porous catalysts and double metal catalytic centers, through which the chaotic radical transport could be reduced in the limited catalytic realms of two-dimensional MOFs, and the generated alkyl hydroperoxide was converted catalytically. These sub-paths listed created a synergistic mechanistic network, and the second and third sub-path is the focus of this study. One innovation of this work was the efficient and selective oxygenation of cycloalkanes and alkyl aromatics with oxygen, which was realized through the synergistic mechanism. In the synergistic mechanism, the usage of the oxygenation intermediate alkyl hydrogen peroxide (R–OOH) as an extra oxidant was achieved, which also reduced the disordered radical diffusion. Based on the above mechanistic consideration, our group is still working on a more efficient catalytic system for hydrocarbon oxygenation with oxygen.

### 3.8. Comparison with Other Catalytic Systems

Finally, the method described here for the cycloalkane and alkyl aromatic oxygenation with oxygen catalyzed by two-dimensional MOFs Co-TCPPNi was compared to the contemporary literature on hydrocarbon oxygenation. As a comparison model, the oxygenation reaction of cyclohexane with oxygen was chosen. As illustrated in Table 8, the primary benefits of the study are (1) no use of toxic solvent, (2) mild reaction conditions, (3) complete product analysis, and (4) higher catalytic performance partly oxygenated under the conditions of (1)~(3). At 120 °C and 1.0 MPa, the selectivity was 92.9%, and the conversion of the substrate was up to 5.6% in the part oxygenation of cyclohexane. The selectivity for partly oxygenated products improved from 85.4% to 92.9% when compared to the metalloporphyrin catalyst T(4-COOCH_3_)PPCo, while the conversion rose from 4.4% to 5.6%. Improving both the conversion and selectivity of partly oxygenated products in C–H bond oxygenation with oxygen was a difficult job since the partly oxygenated products are more reactive than the substrate hydrocarbons, which was realized in this work. Furthermore, GC and HPLC techniques were used to quantify the liquid products with low boiling points and solid products with high boiling points in this study. Therefore, a set of more accurate experimental data was devoted to the cycloalkane and alkyl aromatic oxygenation with oxygen. As a result, our method is a potentially encouraging strategy for improving present manufacturing processes in hydrocarbon oxygenation, as well as a useful reference point for other oxygenation processes involving free-radical species, due to its mild reaction conditions, ease of operation, precise quantification method, simultaneously enhanced conversion, and selectivity.

## 4. Conclusions

Two-dimensional MOFs (M_1_-TCPPM_2_) with microporous lamellar structures and double metal catalytic centers (Co and Ni, Co and Fe, Co and Mn) were successfully synthesized using metalloporphyrins (tetrakis(4-carboxylphenyl)porphyrin metal(II), abbreviated as TCPPM or T(4-COOH)PPM) as linkers in this study. The developed two-dimensional MOF Co-TCPPNi displayed outstanding catalytic performance in the synergistic catalysis of C–H bond oxygenation with oxygen, in which both the conversion rate and selectivity to partly oxygenated products increased with excellent substrate tolerance. The conversion in classical cyclohexane oxygenation increased from 4.4% to 5.6%, and the selectivity of partly oxygenated products increased from 85.4% to 92.9% compared to the sole metal catalyst T(4-COOCH_3_)PPCo. The better catalytic result of Co-TCPPNi was due to the lower reaction temperature, improved oxygenation of hydrocarbons with oxygenation intermediate R–OOH, and the suppression of radical transport using confined catalytic realms of Co-TCPPNi. Furthermore, it was discovered that the oxygenation of hydrocarbons with oxygen primarily proceeded through the free-radical process in a synergistic catalysis model based on the mechanism investigation and comprehensive reaction route study. The method developed in this work avoided the selectivity decrease in partly oxygenated products with the increase in the substrate conversion that existed widely in C–H bond oxygenations with oxygen. Thus, this work provided an attractive and potential example for industrial processes and academic research to solve the selectivity decrease when the conversion rate increased during the partial oxygenation of hydrocarbons with oxygen, and achieved selective and efficient oxygenation of C–H bonds.

## Data Availability

Data available on request due to restrictions privacy.

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
