# Peer review of "Efficient and Selective Oxygenation of Cycloalkanes and Alkyl Aromatics with Oxygen through Synergistic Catalysis of Bimetallic Active Centers in Two-Dimensional Metal-Organic Frameworks Based on Metalloporphyrins"

_biomimetics, 2023, doi:10.3390/biomimetics8030325_

Round 1

Reviewer 1 Report

The reviewed article, titled "Confined Catalytic Realms and Synergistic Catalysis Sites in Two-Dimensional Metal-Organic Frameworks for Highly Selective Oxygenation of Cycloalkanes and Alkyl Aromatics," presents an intriguing study on achieving selective oxygenation of hydrocarbons using bimetallic active centers in two-dimensional metal-organic frameworks (MOFs). The authors have constructed confined catalytic realms and synergistic catalysis sites within the two-dimensional MOFs to facilitate the oxygenation process towards partly oxygenation products.

The authors have performed an extensive set of experiments to investigate the catalytic performance of the bimetallic active centers in the proposed MOFs. The obtained results demonstrate the enhanced efficiency and selectivity achieved through the synergistic catalytic effect. The experimental data presented is thorough and well-documented, allowing for a clear understanding of the observed trends and phenomena. Furthermore, the characterization techniques employed in the study are appropriate and contribute significantly to the credibility of the findings.

While the scientific content of the article is compelling, it is important to address certain language-related aspects to enhance the overall readability and clarity of the paper. The language could benefit from simplifying sentence structures and clarifying technical terms and abbreviations. Additionally, improving paragraph organization and conducting thorough proofreading for grammar and punctuation will contribute to the professionalism and accessibility of the article as follows:

·       Some sentences could be rephrased using simpler language and clearer sentence structures. This would enhance the accessibility of the information to a broader range of readers. Try to avoid duplication.

·       Clarify technical terms and abbreviations: Ensure that all technical terms and abbreviations are properly defined upon their first usage in the text. This will help readers unfamiliar with the specific terminology to follow the discussion more effectively.

·       In addition, one area that requires improvement in the article is the lack of references. While the experimental data and characterization techniques are well-presented, it is important to provide appropriate citations to support the claims and findings made throughout the paper.

·       Enhance paragraph organization: Arrange the content into cohesive paragraphs, each focusing on a specific aspect or finding. This will improve the flow of information and aid in the logical progression of ideas.

·       Proofread for grammar and punctuation: Thoroughly proofread the manuscript to eliminate any grammatical errors or punctuation inconsistencies. Attention to detail in this regard will contribute to the overall professionalism of the article.

In addition, I would like to emphasize that the general comments provided in this review report have been reflected as specific comments in the Word file using track changes. These specific comments aim to provide constructive feedback and suggestions for improvement throughout the manuscript. Authors are encouraged to carefully review the comments and consider the proposed corrections to enhance the clarity, organization, and overall quality of the article. By addressing these specific comments, the authors can further strengthen the scientific content and readability of their work.

Please see the attachment to see more detail comments.

1.     Improve language clarity for better readability and comprehension.

2.     Remove unnecessary duplication and streamline the content.

3.     Revise headlines and subsection titles to be concise and informative.

 These revisions will enhance the overall quality of the paper, making it more polished and impactful.

Reviewer 2 Report

The manuscript entitled “Efficient and selective oxygenation of cycloalkanes and alkyl aromatics with oxygen through synergistic catalysis of bimetallic active centers in two-dimensional metal-organic frameworks based on metalloporphyrins ” by Shen and She et al described the synthesis and characterization of a series of two-dimensional porphyrin-based MOFs, namely M1-TCPPM2, which exhibited outstanding performance in catalyzing highly selective oxygenation of cycloalkanes and alkyl aromatics. This research offered a novel and essential reference for both the efficient and selective oxygenation of C-H bonds and other key chemical reactions involving in free radicals. I believe that this manuscript can be accepted for publication after some minor issues commented on below have been well addressed.

1. The title is too long, suggest to simplify it.
2. The experimental PXRD patterns did not match the simulated one. Is there a reason besides  the different stacking? For an example, the formation of other metal clusters that differ from the binuclear clusters.
3. Some works on porphyrin-based MOFs for selective oxygenation  should be cited, e.g., J. Mater. Chem. A, 2019, 7, 22084.
4. Some works on related porous materials, like COFs, for potential selective oxygenation should be cited, e.g., Molecules, 2022, 27(22), 8002.

Reviewer 3 Report

This manuscript reports oxygenation of cycloalkanes and alkyl aromatic compounds with oxygen catalyzed by bimetallic two-dimensional metal-organic frameworks based on metalloporphyrins. The subject is interesting, and the research is worthwhile. However, there are some issues that should be addressed before the publication of this manuscript:

11)     SEM, XPS and  Nitrogen adsorption-desorption isotherms of all the MOFs mentioned in the manuscript should be provided.

22)    The authors should justify, or at least comment, the low crystallinity (no single crystal diffraction is provided and some PXRD are very poor) and the low values of BET specific surface area.

33) The synergistic catalysis is argued according the following observation: When homogeneous metalloporphyrin T(4-COOCH3)PPCo was replaced by Co-TCPPNi, the conversion in cyclohexane oxygenation was enhanced from 4.4% to 5.6%. This moderate increase on conversion in such low values seems to be insufficient to make this argument. Looking at the series of bimetallic MOFs seems that having Co as M1 is the key for catalytic activity. I suggest control experiments using MOFs with porphyrins without coordinated metal centers.  

4 4)     The mechanistic investigations revealed the involvement of three different radical species derived from cyclohexane. From this observation a complex mechanistic scenario is proposed, which seems too speculative, specially referring to the synergy between Co and Ni centers.

5 5) The values of conversion in this work appear to be lower than most of the published results, as shown in table 8. The authors should make an extra effort to put in value their results.

6 6)  The manuscript is too long. The introduction should be shortened and some of the figures can be transferred to supplementary information.
